# Widespread homogenization of plant communities in the Anthropocene

Barnabas H. Daru [1,2✉], T. Jonathan Davies[3✉], Charles G. Willis[4], Emily K. Meineke[5], Argo Ronk [6], Martin Zobel [7], Meelis Pärtel [7], Alexandre Antonelli [2,8,9,10] & Charles C. Davis [2✉]

Native biodiversity decline and non-native species spread are major features of the Anthropocene. Both processes can drive biotic homogenization by reducing trait and phylogenetic differences in species assemblages between regions, thus diminishing the regional distinctiveness of biotas and likely have negative impacts on key ecosystem functions. However, a global assessment of this phenomenon is lacking. Here, using a dataset of >200,000 plant species, we demonstrate widespread and temporal decreases in species and phylogenetic turnover across grain sizes and spatial extents. The extent of homogenization within major biomes is pronounced and is overwhelmingly explained by non-native species naturalizations. Asia and North America are major sources of non-native species; however, the species they export tend to be phylogenetically close to recipient floras. Australia, the Pacific and Europe, in contrast, contribute fewer species to the global pool of non-natives, but represent a disproportionate amount of phylogenetic diversity. The timeline of most naturalisations coincides with widespread human migration within the last ~500 years, and demonstrates the profound influence humans exert on regional biotas beyond changes in species richness.

[1] Department of Life Sciences, Texas A&M University-Corpus Christi, Corpus Christi, TX 78412, USA. [2] Department of Organismic and Evolutionary Biology, Harvard University Herbaria, Cambridge, MA 02138, USA. [3] Departments of Botany, and Forest & Conservation Sciences, University of British Columbia, Vancouver, BC V6T 1Z4, Canada. [4] Department of Biology Teaching and Learning, University of Minnesota, Minneapolis, MN 55455, USA. [5] Department of Entomology and Nematology, University of California, Davis, CA 95616, USA. [6] Department of Biology, University of Pennsylvania, Philadelphia, PA 19104, USA. [7] Institute of Ecology and Earth Sciences, University of Tartu, Lai 40, EE-51005 Tartu, Estonia. [8] University of Gothenburg and Gothenburg Global Biodiversity Centre, Department of Biological and Environmental Sciences, Carl Skottsbergs gata 22B, SE 405 30 Gothenburg, Sweden. [9] Department of Plant Sciences, University of Oxford, South Parks Road, Oxford OX1 3RB, UK. [10] Royal Botanic Gardens, Kew, Richmond, Surrey TW9 3AE, UK. ✉email: barnabas.daru@tamucc.edu; j.davies@ubc.ca; cdavis@oeb.harvard.edu

Habitat conversion, biotic invasions, anthropogenic climate change, and pollution have contributed initially to dark diversity[1] and eventually to global species losses[2–5]. There has been much focus on species extinctions[6,7]; however, how these biodiversity changes manifest at local to regional scales is still unclear: some studies show declining local diversity[8], while others suggest stable or even increasing species diversity through time[9], and changes in the turnover of species diversity (β-diversity) have been less well studied. Biotic homogenization— declining β-diversity—reduces trait and phylogenetic differences between regions, and is perhaps a more characteristic feature of the Anthropocene than species loss[10]. The Anthropocene epoch encapsulates the profound effects of human activities on the land surface, oceans, atmosphere, and the evolution of life on Earth, with the implications that these changes have no geological analog[10]. Biotic homogenization is primarily driven by native species' extirpation and the introduction and spread of non-native species, commonly due to human activity[11–13]. The balance and influence of these contributing factors remains largely untested across different scales.

For thousands of years, plants have been moved unintentionally or, more commonly, intentionally by humans from their native ranges as sources of food, ornament, medicine, fuel, and shelter[14,15]. Plant invasions greatly accelerated ~500 years ago when the Eastern and Western Hemispheres were united by the Columbian Exchange[16]. One consequence of this widespread movement of species has been the increasing homogenization of plant communities across biomes (e.g.,[17–20]). The magnitude and impact of these compositional changes on the evolutionary structure of native floras across the globe has received surprisingly little attention (see[21,22] for regional assessments). Nonetheless, the importance of evolutionary history in determining the establishment and spread of non-native species has long been recognized. Charles Darwin proposed that introduced species were less likely to establish in communities if they were closely related to the native species—Darwin's naturalization hypothesis[23]. If true, then non-native species would tend to add significantly to the phylogenetic diversity of a region, and reduce phylogenetic turnover between regions. However, if establishment success of non-natives reflects phylogenetically conserved environmental niche preferences (c.f. Darwin's naturalization conundrum)[24], then their addition to the native pool would likely add little to regional phylogenetic diversity, and changes in phylogenetic turnover between regions would be slight[25].

Species extirpations, especially those facilitated by human activities, may also contribute to increasing biotic homogenization of plant communities. While data on past plant extinctions remain sparse (but see[26]), we can extrapolate future extinctions using current Red List assessments[27]. In comparative cross-species analyses, the best predictor of species' extinction risk is geographic range: narrow-ranged endemics, in particular, have the highest risk of extinction[28]. Thus, concomitant with the increasing spread of non-native species, there has been a decline in range-restricted species, which might also contribute to lower rates of species turnover across landscapes. Species extinctions will always result in a loss of evolutionary history, but if extinctions are random across the tree-of-life, then the loss of phylogenetic diversity may be small[29]. There is growing evidence, however, that extinctions tend to be phylogenetically non-random[30], and that species in some clades are at higher risk of extinction[31]. There is as of yet no consensus on expected losses of phylogenetic diversity[31,32]; nonetheless, there is some evidence that evolutionarily distinct plant species might be disproportionately at risk of extinction[33], which could elevate losses[34]. A European study suggested that extinctions increased differentiation of regional floras, but was based on the loss of just

69 species[21]. The aggregate effect of more widespread losses on phylogenetic turnover between regions across the globe have yet to be quantified.

Using a global dataset on ~200,000 vascular plant species, we quantify how non-native naturalizations and recent native extinctions have impacted local (α) and between community (β) plant diversity across spatial scales. We then explore differences in biotic homogenization under varying future scenarios of increasing extinction intensity. By characterizing common routes of human assisted migration we also identify those biomes that are most susceptible to changes in community composition and ecological rearrangement in the Anthropocene[10,35]. We map the distribution of each species using distribution models fitted to carefully curated species occurrence records, and contrast 'Holocene' and 'Anthropocene' species diversities around the globe. We define species composition in the Holocene as the native species' assemblages in each region before widespread migration by humans as initiated by the Columbian Exchange circa 1492[16]. Species composition in the Anthropocene post-date this seminal event, and includes non-native naturalizations, and recent past and projected plant extinctions[26]. However, there is some evidence of non-native plant naturalizations by humans across regions in pre-Columbian times[36,37]. We quantify changes in plant community diversity (α-diversity) between the Holocene and Anthropocene epochs, and examine the signature of increasing homogenization (lower β-diversity) at regional and global scales. We then evaluate the relative contribution of naturalizations vs extirpations in restructuring global plant diversity, and the macroecological correlates of changes in floristic composition across varying extinction scenarios.

Taken together, we reveal that regardless of extinction scenario, the strongest contributor to biotic homogenization results from non-native naturalizations. We show that the biogeographic histories of these recent, human-mediated plant movements between regions is imbalanced. Asia and North America are major sources of non-natives but the species they export are phylogenetically close to recipient floras. These results highlight yet another imprint of the Anthropocene and demonstrate the profound influence humans exert on regional biotas beyond changes in species richness.

## Results and discussion

**Temporal changes in α-diversity across plant communities.** Under a 'best case' scenario defined as recent plant extinctions and naturalizations, but discounting possible future extinctions, we show that the magnitude of naturalizations is far greater than the magnitude of plant extinctions. Approximately 4.9% (10,138) of plant species have been naturalized to a region outside their native ranges (Fig. 1a), while an estimated 0.5% (1065) of species have gone extinct to date (Fig. 1a), leading to an estimated loss of >14,000 million years of evolutionary history (Fig. 1). The trend of declining species and phylogenetic diversity is not an artefact of the spatial resolution (Supplementary Fig. 1), with most losses in North America (particularly California and Florida), Mesoamerica, the Amazon, the Himalaya-Hengduan, Southeast Asia and southwest Australia (Fig. 1d, g, j). These regions are characterized by a number of spectacular clade radiations[38–41], but have also experienced high levels of threat and species invasion[42].

**Temporal changes in compositional turnover across floras.** We found global decreases in β-diversity (the turnover of species and standardized phylogenetic diversity) across most regions (Fig. 2). Shifts towards increasing homogeneity and increasing α-diversity are most pronounced in regions with high elevations and greater rainfall under most scenarios (Supplementary Fig. 2). In

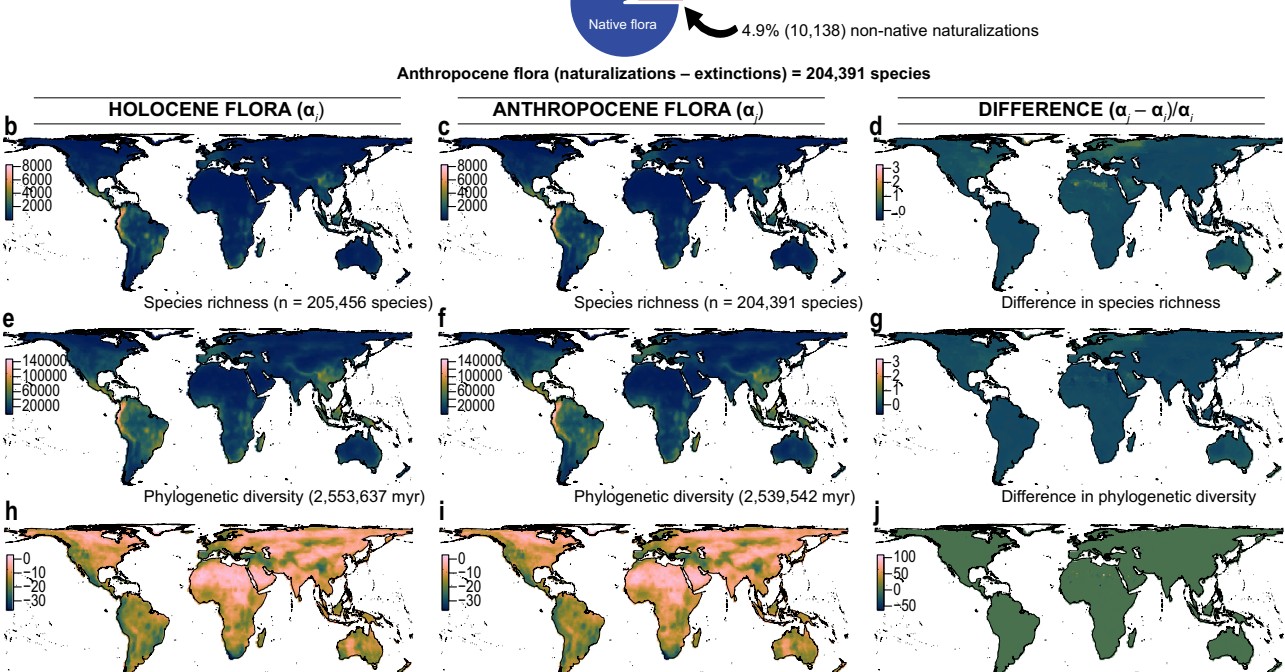

**Fig. 1 Temporal and spatial changes in α-diversity across plant communities in the Anthropocene based on recent plant extinctions and naturalizations (*best case* scenario).** Left panel shows the Holocene flora, middle the Anthropocene flora (based on recent extinctions and naturalizations) and right panel differences between Holocene and Anthropocene floras. **a** Schematic of the Anthropocene flora showing recent extinctions replaced by non-native naturalizations. **b**–**d** Spatial and temporal changes in species (α) diversity. **e**–**g** Spatial and temporal changes in observed phylogenetic (α) diversity. **h**–**j** Spatial and temporal changes in phylogenetic (α) diversity standardized for species richness (phylogenetic tip shuffling 1000 times). Species diversity was calculated as the numbers of species within 100 km × 100 km grid cells (see Supplementary Fig. 1 for a different spatial scale). Phylogenetic diversity (PD) was calculated in million years (myr) as the sum of all phylogenetic branch lengths for the set of species within each grid cell. Species richness was corrected for by calculating the standardized effective size of phylogenetic (α) diversity based on 1000 randomizations (see Methods). Maps are in Behrmann equal-area projection.

particular, Northern Canada, the Saharan Desert (overlapping Chad and Libya), Saudi Arabia, Northern Russia, and Victoria (Australia) are the main epicenters of species and phylogenetic homogenization (Fig. 2c, f).

We demonstrate that changes in α- and β-diversity are driven predominantly by the naturalization of non-native species, rather than recent native species extinctions (Fig. 3). Even under future scenarios of increasing extinction intensity – assuming a 'worst case' when all currently threatened species become extinct – non-native and invasive species naturalizations are by far the strongest contributor to biotic reorganization (Fig. 3). Although our models did not account for non-native colonizations into the future, we suggest that our exploration of alternative extinction scenarios has strong parallels with the widespread use of climate projections to model future ecological scenarios[43–45]. Previous work has indicated that the spread of non-natives might also compensate for biodiversity losses due to species extirpations (e.g.,[9]); however, we find that this is true only to a point, with most regions showing increases in alpha diversity and declines in beta diversity (Fig. 3).

Our results illustrating the disproportionate impact of non-native species are robust to choice of dissimilarity metric (Supplementary Fig. 3), and varying assumptions of extinction pressures (Fig. 3). We also explored whether our results were driven by a few species of large effect—superinvaders—i.e., non-native species with unusually large invaded ranges. While

superinvaders may have a substantial influence on changes in species diversity through time, they account for only 0–14% of the non-native species across all regions, and excluding them does not change our key findings that widespread reorganization of plant communities is primarily due to species naturalizations. The contribution of non-native species to biotic homogenization has been previously documented for birds[46], fish[22,47], insects[48], and plants at regional scales[19,21,22,49,50]. To our knowledge, our study is the first global assessment of plants.

**Exchange of non-native plant species and phylogenetic diversity across continents.** We additionally illustrate how the exchange of species and phylogenetic diversity between regions is strongly asymmetrical (Fig. 4). The exchange and accumulation of non-native plants has been documented previously[15], but their phylogenetic signature has been less well characterized. Here we show that while the principal donors of non-native species are Temperate Asia and North America, the species they export tend to be phylogenetically close to recipient floras (Fig. 4a; Supplementary Tables 2 and 3). Australasia, Pacific and Europe, in contrast, contribute fewer species to the global pool of non-native species, but a disproportionate amount of phylogenetic diversity (Fig. 4b, c). The biased pathways of naturalization we uncover here likely reflect major routes of human-mediated dispersal among regions (perhaps reflecting global trade), the climatic (dis)

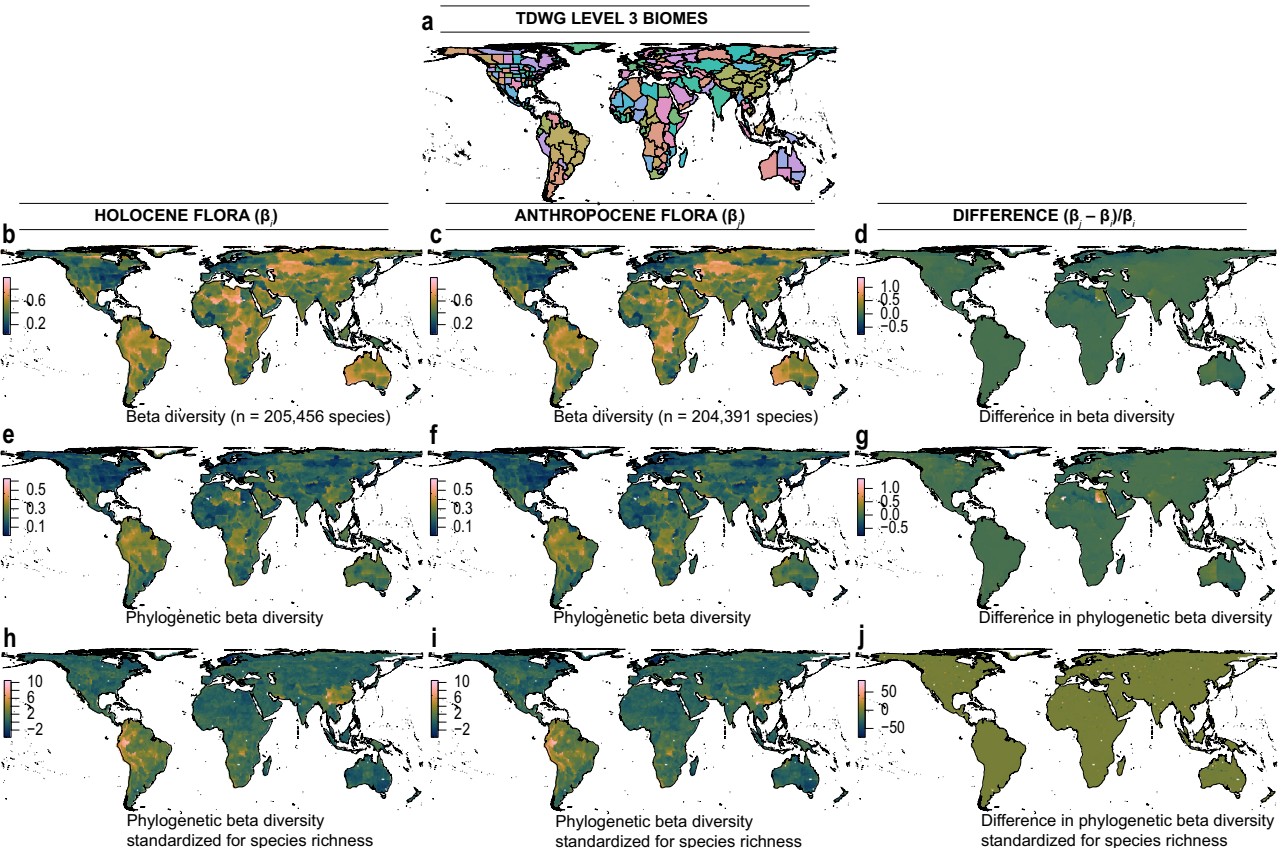

**Fig. 2 Spatial and temporal changes in β-diversity between Holocene (pre-Columbian) and Anthropocene floras based on recent plant extinctions and naturalizations (*best case* scenario).** Left panel shows the Holocene flora, middle the Anthropocene flora (based on recent extinctions and naturalizations) and right panel differences in turnover (homogenization) between Holocene and Anthropocene epochs. **a** The geographic sampling unit within level 3 regional classification as defined by the Biodiversity Information Standards Taxonomic Databases Working Group (TDWG). **b**–**d** Spatial and temporal changes in turnover (β-diversity) in species diversity. **e**–**g** Spatial and temporal changes in turnover (β-diversity) in phylogenetic diversity. **h**–**j** Spatial and temporal changes in phylogenetic β-diversity standardized for species richness (phylogenetic tip shuffling 1000 times). Both species and phylogenetic turnover were calculated using Simpson's metric of beta and phylogenetic beta diversity respectively, between 100 km × 100 km grid cells aggregated across level 3 TDWG biomes, **a**. Maps are in Behrmann equal-area projection.

similarity between donor and recipient regions, and the vulnerability of regional floras to invasion.

**Relationships of non-natives to native flora across spatial scales.** Contrary to Darwin's naturalization hypothesis[23], we find that, on average, non-native species are not phylogenetically distinct from native plant communities (Supplementary Fig. 4). This is not true in all regions (exception Africa, Australasia and Pacific), and superinvaders tend to be more closely related to other non-natives than expected by chance (Supplementary Fig. 4). We also detect strong taxonomic structure in the familial membership of naturalized species. In particular, naturalized species in temperate Asia and North America cluster within similar families ($r = 0.830$; Spearman rank correlation). This is not true in Europe and South America, however, where naturalized species are represented among diverse families ($r = 0.20$; Spearman rank correlation; Supplementary Fig. 5). Such phylogenetic and taxonomic structuring emphasizes the importance of evolutionary history in species naturalization and establishment success, reflecting phylogenetic niche conservatism in environmental preferences and invasive potential[51]. Our analyses at the regional scale thus lend support to the pre-adaptation hypothesis of species invasion, also posited by Darwin[52].

Species are not static in their geographic distributions; some may have been moved by people historically, and today many species are tracking shifting climates. We recognize that generating a reliable estimation of the distribution ranges for extinct species would be challenging plus the historical naturalization of species beyond their native range may have already contributed to the homogenization of local floras. Likewise, data on species naturalizations might be more available than data on species extirpations, potentially biasing us to detect a stronger effect of naturalizations in our analyses. Our analyses thus capture the additional impact on biotic homogenization of more recent anthropogenic activities, and thus likely underestimates the true impact people have on native biodiversity. Further, we believe that as human populations have expanded only relatively recently, historical plant extinctions may have been less likely than historical translocations, and thus our findings that homogenization has been driven largely by naturalizations, rather than extinctions, is likely conservative.

We have demonstrated how recent native species extinctions and, more notably, non-native species naturalizations have reshaped native plant communities across tens of thousands of square kilometers, resulting in profound homogenization of global biodiversity. The floristic shifts we document largely result from human facilitated migrations during the past 500 years (and likely mostly within the past 200 years) and represent yet another imprint of the Anthropocene. Biodiversity change in the Anthropocene often manifests as habitat conversion for human

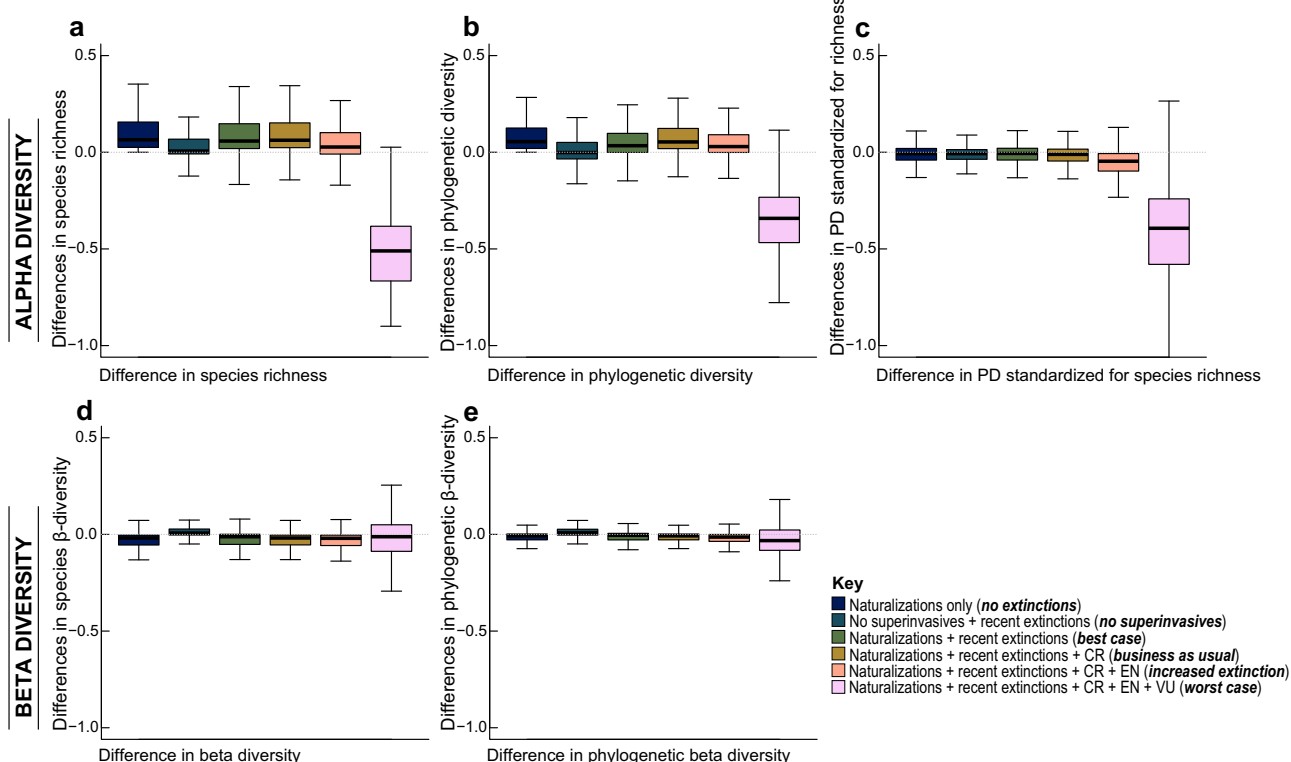

**Fig. 3 Changes in plant communities under various scenarios of extinctions and naturalizations in the Anthropocene.** Top row **a** ($n = 13,218$ grid cells), **b** ($n = 13,218$ grid cells), and **c** ($n = 13,218$ grid cells) shows the differences in α-diversity and bottom row **d** ($n = 13,218$ grid cells), and **e** ($n = 13,218$ grid cells), shows differences in β-diversity. Comparisons are made across six scenarios: i) 'no extinctions' recent naturalizations only, ii) 'no superinvasives' based on the removal of non-native species with unusually large invaded ranges, iii) 'Best case' (based on recent extinctions and naturalizations that have occurred to date), iv) 'business as usual' projected extinction of critically endangered species (CR), v) 'increased extinction' based on projected extinction of endangered (EN) and CR species, and vi) 'worst case' based on projected extinction of all threatened species including vulnerable (VU), EN and CR species. Dashed line at zero corresponds to no change. Species richness was calculated as the numbers of species within 100 km × 100 km grid cells. Phylogenetic diversity was calculated as the sum of all phylogenetic branch lengths for the set of species within each grid cell. The bottom and top of boxes show the first and third quartiles respectively, the median is indicated by the horizontal line, the range of the data by the whiskers. The dataset used for the analysis included 205,456 native species, 1065 recently extinct species, extinction projections for 150,000 species, and 10,138 naturalized species. Source data are provided as a Source Data file.

use, and it is driving the loss of wilderness areas, elevating species extinctions and promoting non-native naturalizations, at scales comparable to a global biodiversity crisis[53,54]. The large-scale transport of species across the globe was likely facilitated initially by long-distance trade and travel via sea, beginning especially with the Columbian Exchange, when food crops, diseases, and populations started to be exchanged between hemispheres by humans[16]. While thousands of species have been spread to new areas unintentionally since this time–for example, as hitchhikers in ship ballast water[55] or dispersed to new climates by migratory animals[56]–others were deliberately introduced to new areas for agriculture and horticulture. The increasing industrialization of agriculture and other drivers of biodiversity change have undoubtedly further accelerated the pace of floristic homogenization within recent decades[57]. The consequences of this global biotic reorganization on ecosystems remain poorly understood, but there is increasing evidence that biotic heterogeneity provides insurance for the maintenance of ecosystem functioning[58] in the face of ongoing global change.

## Methods

**Estimating native plant species' distributions**. We used the newly developed species database, GreenMaps, to estimate native plant species' distributions[59]. GreenMaps includes global distribution maps for ~230,000 vascular plant species. Maps were generated using species distribution models – the statistical estimation of species geographic distributions based on only some known occurrences and

environmental conditions – derived from carefully curated species occurrence records. Occurrence records were obtained from a variety of sources, including herbarium specimens, primary literature, personal observation, and online data repositories including the Global Biodiversity Information Facility[60–62], and Integrated Digitized Biocollections (https://www.idigbio.org/). These records were thoroughly cleaned to reconcile names to follow currently accepted taxonomies [e.g., World Flora Online (www.worldfloraonline.org)], and to remove duplicates and records with doubtful or imprecise localities. Two stringent spatial filters were employed to restrict species' distributions to their known native ranges (i.e., realized niches) and to prevent erroneous records and predictions in areas that contain suitable habitat but are unoccupied by the species (i.e., fundamental niche). First, we applied the spatial constraint, *APGfamilyGeo*, which are expert drawn occurrence polygons ("expert maps") of plant family distributions[63,64] (see Data availability) to restrict species to within these distributions. Second, we applied *GeoEigenvectors*, which are orthogonal variables representing spatial relationships among cells in a grid, encompassing the geometry of the study region at various scales[65]. For the latter, we generated a pairwise geographical connectivity matrix among grid cells to establish a truncation distance for the eigenvector-based spatial filtering, returning a total of 150 spatial filters. These filters were then resampled to the same resolution as the input environmental variables, and were included with the bioclimatic variables in the species distribution modeling. Bioclimatic variables were derived from WorldClim[66] for a total of 19 variables (Supplementary Table 1). Species distribution models (SDMs) were fitted using four different algorithms: generalized linear models (GLM), generalized boosted models (GBM), maximum entropy (MaxEnt), and random forests (RF) with a binomial error distribution (with logit link). Model settings were chosen to yield intermediately complex response surfaces. Model performance was evaluated using area under the receiver operating curve (AUC) and true skill statistic (TSS) scores. AUC scores range from 0 to 1 and should be maximized whereas TSS scores range from −1 to 1. Prior to model building, all predictor variables were standardized. Univariate variable importance for each predictor was assessed in a 5-fold spatial block

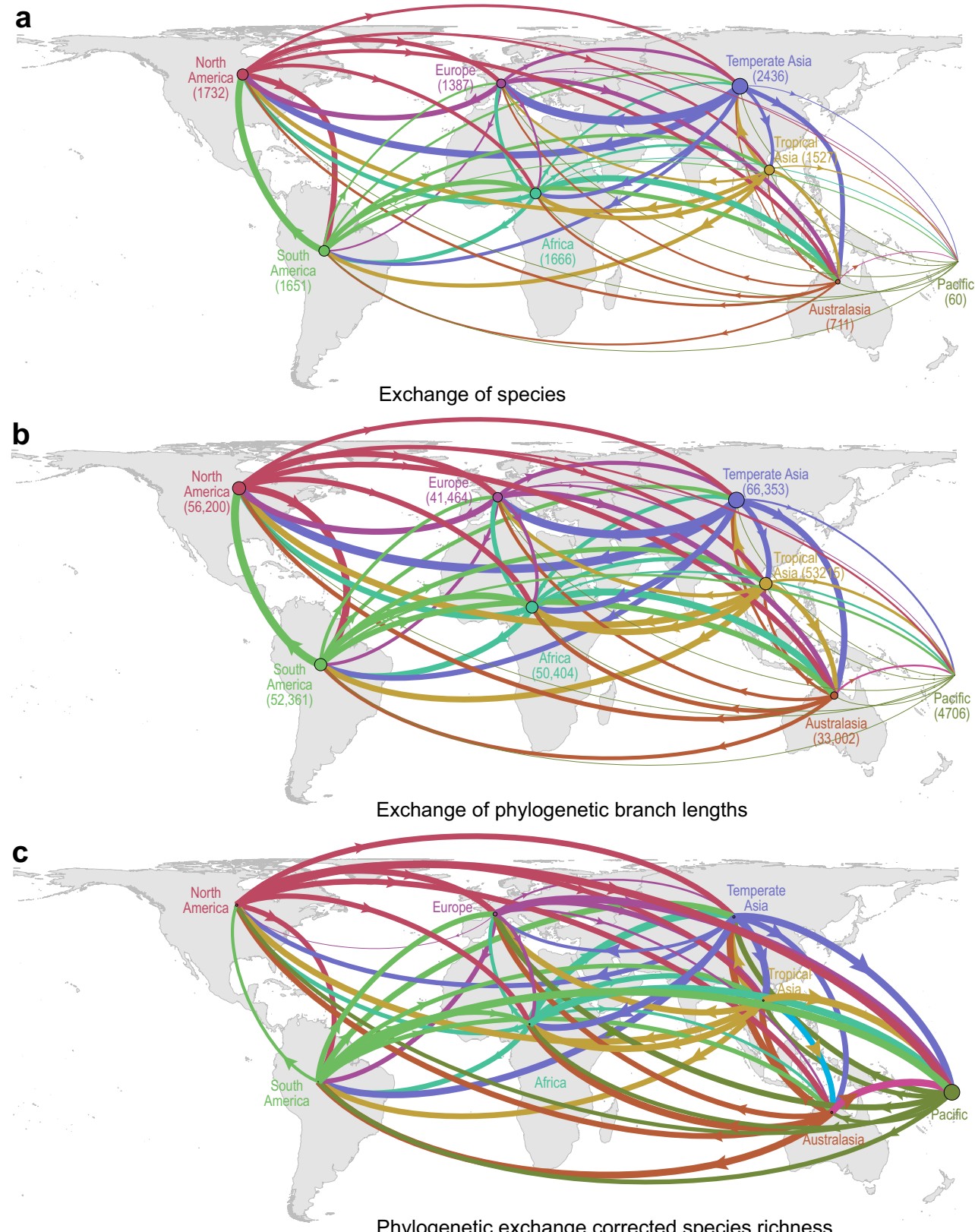

Exchange of species

Exchange of phylogenetic branch lengths

Phylogenetic exchange corrected species richness

cross-validation design. The ensemble predictions from species distribution models were derived using un-weighted ensemble means. Predictive model performance was assessed using a 5-fold spatial block cross-validation. We generated a total of 230,000 range maps, representing species within 382 families at a resolution of 50 × 50 km which was also resampled to 100 × 100 km. To our knowledge, this makes it the largest and only global assessment of geographic distributions for plants at the species-level. Our approach of modeling species distributions follows

the guidelines of ODMAP (Overview, Data, Model, Assessment, Prediction), a comprehensive framework of best practices for reporting species distribution models[67] (see Supplementary Material 1). These maps were stacked and converted to a community matrix for downstream analyses. We also provide a new R function, *sdm*, for performing the SDMs across four algorithms (random forest, generalized linear models, gradient boosted machines, and MaxEnt) tailored for SDMs of large datasets. The *sdm* function is included in our R package

**Fig. 4 Asymmetrical exchange of phylogenetic diversity and non-native plant species across the world. a** Non-native species originating (outbound arrow) or received (inbound arrows) between each continent. Line thickness is proportional to the number of species exchanged. **b** Phylogenetic diversity of non-native species originating (outbound arrow) or received (inbound arrows) between each continent. Line thickness is proportional to the sum of branch lengths exchanged. **c** Net donors and recipients of phylogenetic diversity after correcting for species richness, calculated as the difference in total phylogenetic diversity between the Holocene flora and the Anthropocene flora across continents divided by the number of species exchanged. Arrows indicate the direction of flows from donor to recipient continent, with line thickness proportional to the sum of shared branch lengths weighted by the inverse of species richness. The numbers within parenthesis and circle size represents the number of non-native species or phylogenetic branch lengths in each region. All phylogenetic analyses were run across 100 trees and the median reported. The maps are in Behrmann equal-area projection. A breakdown of nodes and edges exchanged is presented in Supplementary Tables 2 and 3. Source data are provided as a Source Data file.

*phyloregion*[68] along with improved documentation and vignettes to show practical application of this functionality under various modeling scenarios. The *sdm* function was designed with multiple checks such that any species that did not meet one or more checks were filtered out. A feature of novelty of the *sdm* function is the addition of an algorithm that allows a user to exclude records that occur within a certain distance to herbaria, museums or other infrastructure. By default, we used the most updated version of *Index Herbariorum*, a global directory of herbaria[69], but a user has the option to specify their own infrastructure to exclude.

We validated the output distribution maps against the Kew Plants of the World Online database (POWO; http://www.plantsoftheworldonline.org/), which includes native distribution maps for all plants of the world within major biogeographically defined areas recognized by the Biodiversity Information Standards (also known as the Taxonomic Databases Working Group (TDWG))[70]. Although the Kew's distributions of native species are largely based on state/province level such that if a species was observed in any location within a state the whole state is marked as its distribution range, our GreenMaps approach only used the Kew distributions to restrict modeled species distributions within such biogeographic areas. See ref. [59] for full description of the workflow. The range map rasters were converted to a community matrix using the function *raster2comm* in our new R package *phyloregion*[68] for downstream analysis.

**Estimating non-native plant species' distributions**. We used the Global Naturalized Alien Flora (GloNAF) database version 1.2[71,72] to compile a checklist of non-native species, including documented records of alien plants that have dispersed into new regions largely by humans, and which have become successfully naturalized[73,74]. The dataset includes non-native species distributions within TDWG regions. We generated species' distributions for these species using the GreenMaps approach[59] described above, but removing the spatial filters *APGfamilyGeo* and *GeoEigenvectors*. The non-native species ranges were modeled using occurrences that fell outside the boundaries of the native range of each species as determined by Plants of the World Online (POWO). Specifically, we used the following R code to subset occurrences falling outside of POWO as follows:

$$y < -x[!complete.cases(sp :: over(x, powo)), ] \quad (1)$$

where *x* is a data frame of occurrence of a species, and *powo* a shapefile of the native range of the species. We then use the output *y* to model the distribution of non-native species using the *sdm* function in the R package phyloregion[68]. We validated our non-native species distribution models against the GloNAF dataset by overlaying grid cells of non-native species predictions within GloNAF's TDWG levels, and selecting only those projected occurrences that fell within the naturalized range indicated by GloNAF. Such approach allowed us to capture the precise distribution of the non-native species within a state/province as opposed to broadly scoring them present or absent in a state/province as did GloNAF. From our dataset of non-native species, we also identified 'superinvasives', here defined as non-native species with 1.5× the interquartile range above the third quartile of their invaded range size within a TDWG region.

**Recently extinct and threatened plant species**. We compiled information on recent plant extinctions and conservation status of each mapped species. Our dataset of recent extinctions comes from a dataset that includes 1065 plant species that have become extinct since Linnaeus' Species Plantarum[75], derived from a comprehensive literature review and assessments of the International Union for Conservation of Nature (IUCN) Red List of Threatened Species[26,76]. We also explored alternative scenarios of increasing future extinction intensity, considering future losses of currently extant native species, some of which are not currently recognized as of global concern (data from ref. [27]). For the latter analysis, we compiled information on the conservation status of each species and apply the term 'extinction' loosely, which included both native species lost from a region as well as native species that may still be present in some part or all of their native ranges, but they are unlikely to remain so in the near future if current trends continue (see ref. [27]). This dataset comes from machine-learning predictions of conservation status for over 150,000 land plant species[27] defined as the probability of each species as belonging to a Red List non-Least Concern category (i.e., likely of being at risk on some level) based on geographic, environmental, and morphological trait data, variables that are key in predicting conservation risk[27]. For our

purposes here, we assumed that Least Concern species were not at risk of extinction; although we recognize that a substantial proportion of these species may in fact be endangered[27,77]. Within this framework, extinction risk is defined using the expected probability of extinction over 100 years of each taxon[78], scaled as follows: Least Concern = 0.001, Near Threatened and Conservation Dependent = 0.01, Vulnerable = 0.1, Endangered = 0.67, and Critically Endangered = 0.999. We used these statistical projections to estimate future extinction scenarios because they can be fit to over 150,000 land plant species, whereas formal IUCN Red List assessments are currently available for only 33,573 plant species (March 15, 2020).

The final dataset used for our analysis included 205,456 native species, 1065 recently extinct species, extinction projections for 150,000 species, and 10,138 naturalized species.

**Phylogenetic data**. We applied the dated phylogeny for seed plants of the world from ref. [79], which includes 353,185 terminal taxa. The ref. [79] phylogeny was assembled using a hierarchical clustering analysis of DNA sequence data of major seed plant clades and was resolved using data from the Open Tree of Life project. This represents one of the most comprehensive phylogenies of vascular plants at a global scale and includes all species in our analysis. It also provides divergence time estimates to facilitate downstream analytics.

**Data analysis**. We quantified changes in alpha and beta diversity between the Holocene (native species' assemblages in each region before widespread migration by humans as initiated by the Columbian Exchange circa 1492[16]) and Anthropocene (non-native naturalizations, and recent past and projected plant extinctions)[26] epochs across 100 × 100 km grid cells within major biogeographically defined areas recognized by the Biodiversity Information Standards (also known as the Taxonomic Databases Working Group (TDWG))[70]. These TDWG geographic regions correspond to continents, countries, states and provinces. We then explored differences in biotic homogenization under varying future scenarios of extinction including naturalizations only, 'no superinvasives', 'best case' 'business as usual', 'increased extinction' and 'worst case'. Our definition of *best case* refers to recent plant extinctions and naturalizations, and assumes no future extinctions, *business as usual* assumes loss of Critically Endangered (CR) species, *increased extinction* assumes loss of Critically Endangered (CR) and Endangered (EN) species, and the *worst case* scenario assumes loss of all threatened species. Because biodiversity patterns are scale dependent, varying along spatial grains and geographic extents[80,81], we repeated all analyses at spatial grid resolution of 50 × 50 km.

**Temporal changes in α-diversity across plant communities**. For each grid cell, temporal and spatial change in α-diversity was quantified as the difference in species (or phylogenetic) diversity between the Anthropocene (*j*) and Holocene (*i*) periods (see above) expressed as:

$$\Delta\alpha = (\alpha j - \alpha i)/\alpha i \quad (2)$$

Negative Δα values imply that alpha diversity has decreased and positive values indicate increased alpha diversity. Species α-diversity was calculated as the total count of species in each grid cell. Phylogenetic α-diversity was computed as the sum of the phylogenetic branch lengths connecting species from the tip to the root of a dated phylogenetic tree in each grid cell[82]. We also assessed changes in phylogenetic (α) diversity standardized for species richness by calculating standard effects sizes of phylogenetic diversity in communities by shuffling the tips in the phylogeny based on 1000 randomizations. For each iteration of the randomization, the analysis was regenerated using the same set of spatial conditions, but using the randomized version of the tree after which the z-score for each index value was calculated (observed - expected)/sqrt (variance). Temporal changes in α-diversity was assessed at the spatial grain resolution of 50 and 100 km to account for the effects of scale.

**Temporal changes in compositional turnover across floras**. Within TDWG geographic regions, we generated pairwise distance matrices of phylogenetic β-diversity (β$_{phylo}$)[83] and species β-diversity (β$_{tax}$) between all pairs of grid cells, and

compared Holocene and Anthropocene epochs. We used Simpson's index for quantifying compositional turnover because it is insensitive to differences in total diversity among sites[84,85]. The phylogenetic equivalent, $\beta_{phylo}$, represents the proportion of shared phylogenetic branch lengths between cells, and ranges from 0 (species sets are identical and all branch lengths are shared) to 1 (species sets share no phylogenetic branches). We calculated change in compositional turnover ($\Delta\beta$) as:

$$\Delta\beta = (\beta j - \beta i)/\beta i \qquad (3)$$

where $j$ is the Anthropocene species pool and $i$ refers to the Holocene species composition. Negative $\Delta\beta$ values imply that taxonomic/phylogenetic similarity has increased (i.e., biotic homogenization) and positive values indicate biotic differentiation. To assess sensitivity to our choice of diversity index, we re-ran all analyses using Sorensen and Jaccard dissimilarity indices. All (phylogenetic) $\beta$-diversity metrics were calculated using our new R package *phyloregion*[68].

**Effect of superinvasive species**. To determine the extent to which a small number of superinvasive non-native species may be driving patterns of homogenization, we re-ran the analyses described above, but excluded non-native species with the widest ranges within biomes, i.e., species that are more than 1.5× the interquartile range above the third quartile of (invaded) range sizes (i.e., statistical outliers) within TDWG regions. Our definition of range size corresponds to the number of grid cells occupied by a species.

**Phylogenetic structure of naturalizations**. We evaluated whether naturalized species were more likely to have become naturalized in recipient communities in the absence of close relatives—Darwin's naturalization hypothesis—by comparing the mean phylogenetic distance between each non-native species and its nearest phylogenetic neighbor in the recipient flora. Larger mean phylogenetic distances indicate that non-native species tend to be less closely related to the native flora. We first ran each analysis on a set of 100 trees. Significance was assessed by comparing the distribution of observed phylogenetic distances to a null model shuffling non-native status randomly on the tips of the phylogeny (1000 replicates) as implemented in the R package *phyloregion*[68].

**Drivers of change in composition across floras**. To relate change in alpha and beta diversity to possible external drivers, we obtained three sets of variables for each site: (i) ecological: mean annual precipitation (MAP), mean annual temperature (MAT), and elevation; (ii) evolutionary: range size (as proxy for dispersal potential, defined as the average range size across species within a grid cell); and (iii) anthropogenic: wilderness index (inverse of human footprint index). MAP, MAT, and elevation were obtained from the WorldClim database[66]; the geographic range of each species was calculated as the number of cells a species occupied. The Wilderness Index was obtained from ref. [86], and describes the degree to which a place is remote from and undisturbed by the influences of modern society[86]. These variables were converted to Behrmann equal-area projection using the function *projectRaster* in the R package *raster*[87].

We used a linear mixed effects (LME) model of temporal change in, separately, species ($\alpha$) richness, phylogenetic ($\alpha$) diversity, phylogenetic ($\alpha$) diversity standardized for richness, $\beta$-diversity, and phylogenetic $\beta$-diversity between the Anthropocene and Holocene, against ecological, evolutionary and anthropogenic variables as predictors. We used level 3 regions as recognized by the Biodiversity Information Standards as a random effect, allowing us to account for idiosyncratic differences between regions. Changes in metrics of $\beta$-diversity were applied to grid cells by taking the average dissimilarity to other cells within a region as defined by the TDWG level 3 biomes, whereas changes in metrics of $\alpha$ diversity were applied directly to grid cells. We also included a spatial covariate of geographical coordinates as an additional predictor variable to account for spatial autocorrelation. Our model can be formulated as follows:

$$\Delta_i = \beta_0 + \beta_1\,MAT_i + \beta_2\,MAP_i + \beta_3\,elevation_i + \beta_4\,range\_size_i + \beta_5\,wilderness_i + e_i \qquad (4)$$

where $\Delta_i$ is the temporal diversity change (temporal changes in metrics of $\alpha$ or $\beta$ diversity) between the Anthropocene and Holocene in grid cell $i$, $\beta_0$ to $\beta5$ are fixed effect parameters, and $e_i$ is residual error. The LME model was fitted using the lme function in the *nlme* R package[88].

A vignette, with a worked example, data and R codes describing all the steps for the analyses, is also provided on Dryad (https://doi.org/10.5061/dryad.f4qrfj6st).

**Reporting summary**. Further information on research design is available in the Nature Research Reporting Summary linked to this article.

## Data availability

The plant species range maps included in this study come from a newly developed species database called GreenMaps (https://doi.org/10.1101/2020.02.21.960161). GreenMaps includes global distribution maps for ~230,000 vascular plant species. The maps were generated using species distribution models derived from carefully curated species occurrence records, and the dataset is archived on Dryad (https://doi.org/10.5061/dryad.f4qrfj6st). Occurrence records were obtained from a variety of sources, including herbarium specimens, primary literature, personal observation, and online data repositories including the Global Biodiversity Information Facility (Accession: https://doi.org/10.15468/dl.7ujp48; https://doi.org/10.15468/dl.jw4u5a, and https://doi.org/10.15468/dl.m8dzn5), and Integrated Digitized Biocollections (https://www.idigbio.org/). The phylogeny used for the analyses is a published phylogeny that is already available in public repositories. Specifically, the plant phylogeny was downloaded from Smith and Brown (https://doi.org/10.1002/ajb2.1019). Source data are provided with this paper.

## Code availability

All scripts and codes necessary to repeat our analyses have been made available in the Dryad database [https://doi.org/10.5061/dryad.f4qrfj6st] under the folder "R-CODES", and also in the R package *phyloregion*[68].

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

## Acknowledgements

We thank Texas A&M University-Corpus Christi and the Harvard University Herbaria for logistic and financial support. Funding: B.H.D. was supported by the U.S. National Science Foundation (awards 2031928, and 2113424) and Texas Parks and Wildlife (award F21AF03049-00), C.C.D. was supported by the U.S. National Science Foundation (awards 1208835, 1802209, and 1754584), M.Z. and M.P were supported by the University of Tartu (PLTOM20903), Estonian Research Council (PRG609, PRG1065), and by the European Union through the European Regional Development Fund (Centre of Excellence EcolChange).

## Author contributions

The study was conceived and designed by B.H.D. with early development and refinement by C.C.D. All analyses were carried out by B.H.D. The manuscript was written by B.H.D. with significant contributions from T.J.D. and C.C.D. The manuscript was revised by B.H.D., T.J.D., C.G.W., E.K.M., A.R., M.Z., M.P., A.A. and C.C.D. Final approval of the submitted version: B.H.D., T.J.D., C.G.W., E.K.M., A.R., M.Z., M.P., A.A. and C.C.D.

## Competing interests

The authors declare no competing interests.
