## [Peer Review File · Nature Communications]

Peer Review comments, initial round review –

Reviewer #1 (Remarks to the Author):

This is definitively a well written and structured paper. I am feeling there are the following points to be improved during the next review round:

- DARK DIVERSITY: Some of the authors of this manuscript worked hard on dark diversity in the past. I am feeling that this issue is not considered at all in this manuscript while it might be appropriate, especially once attempting at estimating changes in alpha and beta diversity at different spatial scales. A paragraph might be added. How dark diversity would impact such estimates?
- HUMAN IMPACT: the human impact on species distributions is rapidly mentioned through the manuscript. I am feeling that in a manuscript including "anthropocene" in the title, this should be expanded.
- TDWG LEVEL 3 BIOMES: Might there be any area effects of TDWG LEVEL 3 BIOMES on the estimates of diversity for both the holocene and anthropocene floras?
- MISUSE OF COLOURS (Cramieri et al., Nature Comm.): There has been a long debate, even in Nature Communications, about the misuse of colours in scientific mapping (e.g. in Nature Communications: Cramieri et al. The misuse of colour in science communication). This is the occasion to follow their suggestion and make a color ramp palette against colour blindness. At the time being you are using blue to red which is a problem for a lot of eye diseases. This is a critical point about properly mapping diversity.
- FIGURE 4: Figure 4 is impressive and the related code might be shared with other researchers.

Minor points:

In all the figures, please better arrange text, to be exactly in line and in column.

Compliments for this nice paper!

Duccio Rocchini

--

Duccio Rocchini, PhD

Full Professor

Alma Mater Studiorum University of Bologna

<https://www.unibo.it/sitoweb/duccio.rocchini/en/>

Reviewer #2 (Remarks to the Author):

I think this is a really exciting study, and one I expect will feature prominently in the next State of the World's Plants publication. Unfortunately, I feel that there are some fundamental flaws that may

preclude its publication in the current state (at least in a journal of this level of prestige).

Main Comments

1) This paper seems to critically depend on a resource that has not been peer reviewed, GreenMaps. After digging into this publication and the underlying R scripts, it looks as though a relatively default approach to maxent modelling was used, may be problematic. It is unclear the extent to which species-specific tuning was done. I think this could be greatly clarified through the use of one of the recently developed protocols for reporting SDM decisions, e.g.

Merow, C, Maitner, BS, Owens, HL, et al. Species' range model metadata standards: RMMS. *Global Ecol Biogeogr.* 2019; 28: 1912– 1924. <https://doi.org/10.1111/geb.12993>,

Feng, X., Park, D.S., Walker, C. et al. A checklist for maximizing reproducibility of ecological niche models. *Nat Ecol Evol* 3, 1382–1395 (2019). <https://doi.org/10.1038/s41559-019-0972-5>

Zurell et al. 2020:

<https://doi.org/10.1111/ecog.04960>

2) It seems like it may be a bit of a missed opportunity to use POTW to delineate native species, but only GLONAF to delineate introduced species. If we accept the POTW checklist as true, then any records not labelled as introduced in POTW must be non-native. Presumably this would give you a larger number of introduced species that could be modelled. Comparing this set of species known to be introduced with those known to be naturalized may help you put confidence intervals on the number of invasives in any region. The true number should be between these estimates and if both show the same pattern I think that would make for a very strong argument that these results are pretty robust.

3) The phylogeny used here is full of polytomies. This will bias the various phylogenetic metrics calculated. I think a reasonable solution to this is randomly resolving the phylogenies (or resolving them according to some model of evolution). This would eliminate biases due to polytomies and would also allow you to say something about how phylogenetic uncertainty influences your results.

4) Darwin's invasion hypotheses deal with the likelihood that an introduced species becomes established, although I've also read arguments that they can be applied to rate of spread. Thus, a strong test of the hypothesis requires that we know something about failed introductions. Perhaps you could get at this a bit using something like I mention in point 2 above. Alternatively, perhaps rephrasing this to acknowledge the shortcomings and assumptions of the present approach might work.

Minor Comments

106-107: Are these 4.3 percent introduced, or is this restricted to naturalized species?

123-126: I think its important to be clear with the wording: these species haven't been assessed as "threatened", they're inferred to be threatened based on a model. I think this is reasonable to do, its just important to be more clear with the phrasing

364-368: These data aggregators should be cited. E.g. GBIF provides download citations.

368-370: How were the names reconciled? The use of e.g. seems to imply you used multiple taxonomies, but you only list one. If more were listed, it should be made explicit which ones were used. Was this manual cleaning, or did you use some method of automation (e.g. fuzzy matching?)

Were records only excluded if they were exact duplicates, or simply identical coordinates? How do you detect uncertain or imprecise coordinates? Did you exclude other common problems (e.g. herbaria, centroids, etc.?)

374: Are the family maps you used included with GreenMaps or published elsewhere online? If not, would be useful to provide these in the SI for reproducibility.

381: It would be good to specify what the 19 bioclim layers are for those who are unfamiliar.

386: Were maps discarded if they performed poorly?

399-402: I think this is a bit of an overstatement and a bit misleading as phrased. These factors do indeed influence the models, but I think its more appropriate to say these are all things that influence the model (some of which might bias it, e.g. sampling), than things that are captured by it.

410-412: So, for the non-native ranges, were the maps built with ONLY points in the introduced region, or with point in both native and introduced regions? Did you restrict the occurrences to only those that fell within the regions specified by glonaf?

483: Unfortunately, recent work by Brody Sandel suggests that this is insufficient to standardize for richness and it is also necessary to rarify the number of species to a common value and then examine things across many rarified samples.

Reviewer #3 (Remarks to the Author):

Comments for the authors

The manuscript entitled “Widespread homogenization of plant communities in the Anthropocene” is based on a study assessing the effects of plant species extinctions and non-native plant species introductions on the homogenization of plant communities globally. To this end the authors model the distribution of more than 200,000 native plant species and nearly 9,000 non-native introduced species, and calculate global diversity indexes under different scenarios. The authors find that non-native species introductions are much more important as drivers of biotic homogenization, compared with species extinctions. This type of studies clearly make a big contribution to our understanding of the impacts of humans on biodiversity. However, I have some concerns that I consider that the authors should address:

- If data on plant species occurrences is not enough to model the distribution for most plant species (as mentioned in lines 139-140 in Daru (2020)), then how can the authors be sure that their analyses based on species distributions are reliable?

Daru, B. H. (2020) GreenMaps: a tool for addressing the Wallacean shortfall in the global distribution of plants, bioRxiv 2020.02.21.960161

- Data on species introductions is likely to be more thorough (showing values closer to reality) than on species extinctions (even while only focusing on recent extinctions), potentially introducing a bias in the analyses presented here. As a result, the authors may be underestimating the effect of species extinctions. I consider that the authors should at least discuss this possibility. This is especially important considering that the authors conclude that the effect of species extinctions is insignificant compared to the effect of non-native species introductions as drivers of biotic homogenization.

- If the authors used GloNAF as a checklist for introduced plant species how is it possible that according to the authors there are 8,839 plant species that have been introduced to a region outside

their native range, while GloNAF includes more than 12,000 species (i.e. without considering subspecies and varieties) that have already become naturalized outside their native range? Please explain (Lines 106-107)

- Using 1492 as a cut-off date for what is native and what is not native is standard, but may not be correct. There are lots of evidence of pre 1492 movement of species by humans (in Eurasia and South America, are relatively well studied). I suggest the authors make a clearer case on this.

Specific comments with line numbers:

Why are there no section titles (Introduction, Results, Discussion)?

Line 54: Please add the following citation:

van Kleunen M, Xu X et al. (2020) Economic use of plants is key to their naturalization success. *Nature Communications* 11:3201

Lines 133-134: How can you assess the disproportionate impact of non-native species across dissimilarity metrics from the maps on extended data Figure 3?

Lines 137-140: Where are the results from analyses excluding super invaders?

Lines 147-149: According to Extended data table 2 the principal donors of non-native species are Temperate Asia and North America (not South America).

Lines 175-177: However, data on species introductions is likely to be more thorough than on species extinctions, potentially introducing a bias in your analyses. As a result, you may be underestimating the effect of species extinctions.

Line 355: If circles in Figure 4 represent the number of native species in each region does this mean that (according to Figure 4c) the number of native plant species in Europe is notoriously higher than the number of native plant species in South America (including the Amazon rainforest) or in Africa (including the Congo rainforest)? This does not seem to agree with Figure 2 in Daru (2020).

Daru, B. H. (2020) GreenMaps: a tool for addressing the Wallacean shortfall in the global distribution of plants, *bioRxiv* 2020.02.21.960161

Line 409: I think Richardson et al. (2000b) is a more appropriate citation than the current citation (Richardson et al. 2000a):

Richardson DM, Allsopp N, D'Antonio CM, Milton SJ, Rejmanek M (2000a) Plant invasions—the role of mutualisms. *Biological Reviews* 75:65-93

Richardson DM, Pyšek P, Rejmánek M, Barbour MG, Panetta FD, West CJ (2000b) Naturalization and invasion of alien plants: concepts and definitions. *Diversity and Distributions* 6:93-107

Line 467: There is a comma missing after “best case”

REVIEWER COMMENTS

Reviewer #1 (Remarks to the Author):

This is definitively a well written and structured paper. I am feeling there are the following points to be improved during the next review round:

- DARK DIVERSITY: Some of the authors of this manuscript worked hard on dark diversity in the past. I am feeling that this issue is not considered at all in this manuscript while it might be appropriate, especially once attempting at estimating changes in alpha and beta diversity at different spatial scales. A paragraph might be added. How dark diversity would impact such estimates?

RESPONSE 1.1: We have now added discussion about dark diversity in the Introduction in Lines 43-44 as follows (underlined for emphasis):

“Habitat conversion, biotic invasions, anthropogenic climate change, and pollution have contributed initially to dark diversity¹ and eventually to global species losses²⁻⁵”

- HUMAN IMPACT: the human impact on species distributions is rapidly mentioned through the manuscript. I am feeling that in a manuscript including "anthropocene" in the title, this should be expanded.

RESPONSE 1.2: We have added discussion about human impact in the Anthropocene in the Introduction and Discussion as follows:

Introduction (Lines 50-53):

“The Anthropocene epoch encapsulates the profound effects of human activities on the land surface, oceans, atmosphere, and the evolution of life on Earth, with the implications that these changes have no geological analogue¹⁰.”

Discussion (Lines 198-200):

“Biodiversity change in the Anthropocene often manifests as habitat conversion for human use, and it is driving the loss of wilderness areas, elevating species extinctions and promoting non-native introductions, at scales comparable to a global biodiversity crisis^{53,54}.”

- TDWG LEVEL 3 BIOMES: Might there be any area effects of TDWG LEVEL 3 BIOMES on the estimates of diversity for both the holocene and anthropocene floras?

RESPONSE 1.3: This is an important point. The Taxonomic Diversity Working Group (TDWG), also known as the World Geographical Scheme for Recording Plant Distributions, provides an agreed system of geographical units at approximately "country" level and higher for recording plant distributions. We tested the effects of spatial scale by running analysis at different spatial extents of the TDWG biomes at level 1 (corresponding to continental extent), level 2 (regional extent) and level 3 (country extent), and have incorporated the results in the supplementary information. For example, analyses of Darwin's naturalization hypothesis was done across three spatial extents of the TDWG levels 2, 3, and 4 (See Supplementary Figure 4 below), whereas temporal changes in alpha diversity were analyzed at varying spatial grain sizes of 50 km × 50 km and 100 km × 100 km grid cells (Supplementary Figure 1 below). We did not find any

qualitative differences in results to those we report in the main text, suggesting that our findings are robust across varying spatial extents and grain sizes.

Supplementary Figure 4 | Test of Darwin’s naturalization hypothesis. Comparison of the phylogenetic relatedness of superinvasives and other non-natives to the native species pools across different spatial scales based on TDWG biomes (Biodiversity Information Standards Taxonomic Databases Working Group). **a**, Phylogenetic relatedness calculated for level 2 of TDWG biomes corresponding to the spatial extent of regions (subcontinents). **b**, Phylogenetic relatedness calculated for level 3 of TDWG biomes corresponding to "botanical countries" (which often ignore purely political considerations). **c**, Phylogenetic relatedness calculated for plants at level 4 of TDWG biomes corresponding to "basic recording units" where political integrity is recognized. Results show standard effect size of mean phylogenetic distance estimated from 1000 randomizations. Significance was assessed as the lack of overlap between the 95% confidence interval and zero. An overlap on the other hand indicates non significance but there might be some exceptions. Temp Temperate, Trop Tropical, Au Australasia, Eu Europe, N Am Northern America, Pac Pacific, S Am Southern America.

Supplementary Figure 1 | Temporal and spatial changes in α -diversity across plant communities in the Anthropocene based on recent plant extinctions and introductions (best case scenario) at a spatial resolution of 50 km \times 50 km. Left panel shows the Holocene flora, middle the Anthropocene flora (based on recent extinctions and introductions) and right panel differences between Holocene and Anthropocene floras. (a) Schematic of the Anthropocene flora showing recent extinctions replaced by non-native introductions. (b), (c), (d) Spatial and temporal changes in species (α) diversity. (e), (f), (g) Spatial and temporal changes in observed phylogenetic (α) diversity. (h), (i), (j) Spatial and temporal changes in phylogenetic (α) diversity standardized for species richness (phylogenetic tip shuffling 1000 times). Species diversity was calculated as the numbers of species within 50 km \times 50 km grid cells. Phylogenetic diversity (PD) was calculated in million years (myr) as the sum of all phylogenetic branch lengths for the set of species within each grid cell. Species richness was corrected for by calculating the standardized effective size of phylogenetic (α) diversity based on 1000 randomizations. Maps are in Behrmann equal-area projection.

- MISUSE OF COLOURS (Cramieri et al., Nature Comm.): There has been a long debate, even in Nature Communications, about the misuse of colours in scientific mapping (e.g. in Nature Communications: Cramieri et al. The misuse of colour in science communication). This is the occasion to follow their suggestion and make a color ramp palette against colour blindness. At the time being you are using blue to red which is a problem for a lot of eye diseases. This is a critical point about properly mapping diversity.

RESPONSE 1.4: Excellent suggestion. We have now revised all our maps to use scientific colour ramps that account for colour vision deficiency, for example, by using the “batlow” colour palette in the R package scico (Pedersen and Cramer 2020). We feel that these changes result in a more effective presentation of the potential and novelty of the findings. Specifically, all the

heat maps in Figures 1, 2, Supplementary Figs 1, and 3 have been revised to use the “batlow” colour palette in the R package scico (Pedersen and Cramer 2020). This information is now clearly provided in the revised manuscript (in Lines 334, 348, 688) and revised Figures (see example of Figure 1 below):

Fig. 1 Temporal and spatial changes in α -diversity across plant communities in the Anthropocene based on recent plant extinctions and introductions (best case scenario). Left panel shows the Holocene flora, middle the Anthropocene flora (based on recent extinctions and introductions) and right panel differences between Holocene and Anthropocene floras. **a**, Schematic of the Anthropocene flora showing recent extinctions replaced by non-native introductions. **b, c, d**, Spatial and temporal changes in species (α) diversity. **e, f, g**, Spatial and temporal changes in observed phylogenetic (α) diversity. **h, i, j**, Spatial and temporal changes in phylogenetic (α) diversity standardized for species richness (phylogenetic tip shuffling 1000 times). Species diversity was calculated as the numbers of species within 100 km \times 100 km grid cells (see Supplementary Fig. 1 for a different spatial scale). Phylogenetic diversity (PD) was calculated in million years (myr) as the sum of all phylogenetic branch lengths for the set of species within each grid cell. Species richness was corrected for by calculating the standardized effective size of phylogenetic (α) diversity based on 1000 randomizations (see Methods). Maps are in Behrmann equal-area projection.

- FIGURE 4: Figure 4 is impressive and the related code might be shared with other researchers.

RESPONSE 1.5: In this revision, we now provide the R codes in the supplementary information and via our R package’s website (<https://phyloregion.com/>), on how Figure 4 was generated. In addition, we discuss how we have dealt with this below in response to another reviewer’s similar comment where we revised the figure using an updated dataset. This revised figure is now provided in Line 373 as follows:

Fig. 4 Asymmetrical exchange of phylogenetic diversity and non-native plant species across the world. a, Non-native species originating (outbound arrow) or received (inbound arrows) between each continent. Line thickness is proportional to the number of species exchanged. **b**, Phylogenetic diversity of non-native species originating (outbound arrow) or received (inbound arrows) between each continent. Line thickness is proportional to the sum of branch lengths exchanged. **c**, Net donors and recipients of phylogenetic diversity after correcting for species richness, calculated as the difference in total phylogenetic diversity between the Holocene flora and the Anthropocene flora across continents divided by the

number of species exchanged. Arrows indicate the direction of flows from donor to recipient continent, with line thickness proportional to the sum of shared branch lengths weighted by the inverse of species richness. The numbers within parenthesis and circle size represents the number of non-native species or phylogenetic branch lengths in each region. All phylogenetic analyses were ran across 100 trees and obtained a median result. The maps are in Behrmann equal-area projection. A breakdown of nodes and edges exchanged is presented in Supplementary Tables 2–3.

Minor points:

In all the figures, please better arrange text, to be exactly in line and in column.

RESPONSE 1.6: Done. We have revised all figures and text to be exactly in line and in column by using the “ruler” tool of our image visualization software. An example of how we have achieved this is presented in Figure 2 as shown below:

Fig. 2 Spatial and temporal changes in β -diversity between Holocene (pre-Columbian) and Anthropocene floras based on recent plant extinctions and introductions (best case scenario). Left panel shows the Holocene flora, middle the Anthropocene flora (based on recent extinctions and introductions) and right panel differences in turnover (homogenization) between Holocene and Anthropocene epochs. **a**, The geographic sampling unit within level 3 regional classification as defined by the Biodiversity Information Standards Taxonomic Databases Working Group (TDWG). **b, c, d**, Spatial and temporal changes in turnover (β -diversity) in species diversity. **e, f, g** Spatial and temporal changes in turnover (β -diversity) in phylogenetic diversity. **h, i, j** Spatial and temporal changes in phylogenetic β -diversity standardized for species richness (phylogenetic tip shuffling 1000 times). Both species and

phylogenetic turnover were calculated using Simpson's metric of beta and phylogenetic beta diversity respectively, between 100 km × 100 km grid cells aggregated across level 3 TDWG biomes, **a**. All analyses of phylogenetic β -diversity were based on a randomly selected subset of 100 trees from a random distribution of 1000 trees. Maps are in Behrmann equal-area projection.

Compliments for this nice paper!

Duccio Rocchini

RESPONSE 1.7: We thank the Reviewer for the nice and positive remarks on our manuscript.

--

Duccio Rocchini, PhD

Full Professor

Alma Mater Studiorum University of Bologna

<https://www.unibo.it/sitoweb/duccio.rocchini/en/>

Reviewer #2 (Remarks to the Author):

I think this is a really exciting study, and one I expect will feature prominently in the next State of the World's Plants publication. Unfortunately, I feel that there are some fundamental flaws that may preclude its publication in the current state (at least in a journal of this level of prestige).

RESPONSE 2.1: We appreciate the detailed and constructive comments from the reviewer. We have made significant updates on the underlying *GreenMaps* data by generating new maps using an ensemble modeling based on four different algorithms (random forest, generalized linear models, gradient boosted machines, and MaxEnt). We have also made a number of significant changes in the manuscript. We feel that these changes result in a more effective communication of the key finding that nonnative species drive widespread homogenization of plant communities.

Main Comments

1) This paper seems to critically depend on a resource that has not been peer reviewed, *GreenMaps*. After digging into this publication and the underlying R scripts, it looks as though a relatively default approach to maxent modelling was used, may be problematic. It is unclear the extent to which species-specific tuning was done. I think this could be greatly clarified through the use of one of the recently developed protocols for reporting SDM decisions, e.g.

Merow, C, Maitner, BS, Owens, HL, et al. Species' range model metadata standards: RMMS. *Global Ecol Biogeogr*. 2019; 28: 1912– 1924. <https://doi.org/10.1111/geb.12993>,

Feng, X., Park, D.S., Walker, C. et al. A checklist for maximizing reproducibility of ecological niche models. *Nat Ecol Evol* 3, 1382–1395 (2019). <https://doi.org/10.1038/s41559-019-0972-5>

Zurell et al. 2020:

<https://doi.org/10.1111/ecog.04960>

RESPONSE 2.2: These are important points and so we structure our responses accordingly as follows:

1) While the early version of GreenMaps (v.1.1) was based only on MaxEnt modeling, we have now expanded GreenMaps to average these models over four algorithms (random forest, generalized linear models, gradient boosted machines, and MaxEnt) to generate ensemble species distributions model (SDM) predictions. In addition, our approach saves as output the predictions of the relative probability of presence, input occurrence data, and performance statistics. We now provide a new R function, *sdm*, for performing the SDMs across four algorithms (random forest, generalized linear models, gradient boosted machines, and MaxEnt) tailored for SDMs of large datasets such as the over 230,000 plant species at a global scale. The *sdm* function is now included in our R package *phyloregion* (Daru et al. 2020) along with improved documentation and vignettes to show practical application of this functionality under various modelling scenarios. A feature of novelty of the *sdm* function is the addition of an algorithm that allows a user to exclude records that occur within a certain distance to herbaria, museums or other infrastructure. By default, we used the most updated version of *Index Herbariorum*, a global directory of herbaria (Thiers 2016), but a user has the option to specify their own infrastructure to exclude. We have made a package version update where these products are now all freely available via the Comprehensive R Archive Network (CRAN) and GitHub.

2) We have now described how the updated version of GreenMaps (v.1.2) substantially borrows from established approaches such as the ODMAP (Overview, Data, Model, Assessment and Prediction) protocol of Zurell et al. (2020), and thus provides reliable and reproducible outputs, which underpin our analyses and support the results in our study. We have also included in the supplementary information a refined protocol tailored for reporting the GreenMaps species' distribution models based on the ODMAP protocol of Zurell et al. (2020).

3) Using the refined protocol, we regenerated the SDMs using parallel computing nodes on the high performance computing clusters of Texas A&M University-Corpus Christi and Harvard University, resulting in SDMs for a total of 226,806 species based on ensembled predictions. Thus, all analyses reported in this revised manuscript are based on the updated GreenMaps data version 1.2. All data, scripts and codes necessary to repeat our analyses have been made available at the Dryad data repository along with a well annotated README file on how to use the data (<https://datadryad.org/stash/share/UFhi3ts7G6sljHj1IAnUUK1V8AVzh4ep6hEUdqCJV9k>).

4) We are currently developing an R *shiny* application for GreenMaps to be available as a permanent resource that will allow users to access the data more widely, do their own exploratory data analysis and visualize biogeographic patterns and distribution maps as well as download data. The GreenMaps shiny application is at 30% completion. Below is a simple wireframe for GreenMaps *shiny* application.

Supplementary data 1. A simple wireframe of GreenMaps shiny application showing two tabs. a Single species maps, and b biodiversity maps.

2) It seems like it may be a bit of a missed opportunity to use POTW to delineate native species, but only GLONAF to delineate introduced species. If we accept the POTW checklist as true, then any records not labelled as introduced in POTW must be non-native. Presumably this would give you a larger number of introduced species that could be modelled. Comparing this set of species known to be introduced with those known to be naturalized may help you put confidence intervals on the number of invasives in any region. The true number should be between these estimates and if both show the same pattern I think that would make for a very strong argument that these results are pretty robust.

RESPONSE 2.3: This is a great point. We agree that it is of value to explore non-native ranges based on species whose ranges fall outside the boundaries of Plants of the World Online (POWO) as input to model the distributions of non-natives, and that this would increase confidence in our results. We pursued this suggestion by running the analysis using our new R function *sdm* (now incorporated in our phyloregion R package) which has built-in option to restrict predictions within or outside a given boundary, in this case the boundaries of the POWO. Using this approach, we were able to generate range maps for 10,138 non-native species compared to 8839 previously. All the analyses reported in the revised manuscript are based on the new data. Results showed substantial congruence to previous approach. This information is now provided in the Methods in Lines 455-462 as follows:

“The non-native species ranges were modeled using occurrences that fell outside the boundaries of the native range of each species as determined by Plants of the World Online (POWO). Specifically, we used the following R code to subset occurrences falling outside of POWO as follows:

```
y <- x[!complete.cases(sp::over(x, powo)),]
```

where x is a data frame of occurrence of a species, and powo a shapefile of the native range of the species. We then used the output y to model the distribution of non-native species using the sdm function in the R package phyloregion⁶⁷.”

3) The phylogeny used here is full of polytomies. This will bias the various phylogenetic metrics calculated. I think a reasonable solution to this is randomly resolving the phylogenies (or resolving them according to some model of evolution). This would eliminate biases due to polytomies and would also allow you to say something about how phylogenetic uncertainty influences your results.

RESPONSE 2.4: We acknowledge the presence of polytomies in the phylogenetic data that could influence the results. We first attempted to resolved polytomies in the phylogenetic tree by using the *multi2di* function in the R package *ape* (Paradis and Schliep 2019) to transform all multichotomies into a series of dichotomies, which resulted in 100 randomly resolved trees (see R code below). However, rerunning the phylogenetic analysis across 100 distribution of trees and obtaining a median effectively gave identical results to that of the original tree. Hence, we reported our results based on the original tree from Smith and Brown (2018), with the expectation that this should a minor effect on the metrics we calculated. The R code we tried for resolving the tree is as follows:

```
library(ape)
phy <- read.tree("path_to_file/tree.tre")
resolved <- replicate(100, multi2di(phy), simplify = FALSE)
class(resolved) <- "multiPhylo"
```

4) Darwin's invasion hypotheses deal with the likelihood that an introduced species becomes established, although I've also read arguments that they can be applied to rate of spread. Thus, a strong test of the hypothesis requires that we know something about failed introductions. Perhaps you could get at this a bit using something like I mention in point 2 above. Alternatively, perhaps rephrasing this to acknowledge the shortcomings and assumptions of the present approach might work.

RESPONSE 2.5: The reviewer is correct. The topic of Darwin's naturalization hypothesis or conundrum is very broad, with different views, all of which vary with taxonomic groups and across scales. Although we do not attempt a comprehensive review of this large field, and we would not consider ourselves expert enough to do this effort justice, we here revised our manuscript to evaluate whether introduced species were more likely to have become naturalized in recipient communities in the absence of close relatives. We achieved this in two ways: 1) comparison of the mean phylogenetic distance between each non-native species and its nearest phylogenetic neighbor in the recipient flora, and 2) by comparing ranks for plant families containing non-native species interchanged across continents. We hope that this results in a more effective presentation of the potential value and novelty of our paper – non-native species are not necessarily phylogenetically distinct from native plant communities at the scale of our analysis, and that there is a strong taxonomic structure in the familial membership of introduced species – which we believe are major contributions of this revised manuscript.

This information is now provided in Lines 166-179 as follows:

“Relationships of non-natives to native flora across spatial scales. Contrary to Darwin's naturalization hypothesis²³, we find that, on average, non-native species are

not phylogenetically distinct from native plant communities (Supplementary Fig. 4), but not in all regions (exception Africa, Australasia and Pacific, and superinvaders tend to be more closely related to other non-natives than expected by chance (Supplementary Fig. 4). We also detect strong taxonomic structure in the familial membership of introduced species. In particular, introduced species in temperate Asia and North America cluster within similar families ($r = 0.830$; Spearman rank correlation). This is not true in Europe and South America, however, where introduced species are represented among diverse families ($r = 0.20$; Spearman rank correlation; Supplementary Fig. 5). Such phylogenetic and taxonomic structuring emphasizes the importance of evolutionary history in species introduction and establishment success, reflecting phylogenetic niche conservatism in environmental preferences and invasive potential⁵¹. Our analyses at the regional scale thus lend support to the pre-adaptation hypothesis of species invasion, also posited by Darwin⁵².”

Minor Comments

106-107: Are these 4.3 percent introduced, or is this restricted to naturalized species?

RESPONSE 2.6: We use the terms ‘introduced’ and ‘non-native’ interchangeably as alien plants dispersed into new regions due to human activities and successfully naturalized. For added clarity, the wording of the sentence has been revised as follows:

“Approximately 4.9% (10,138) of plant species have been introduced to a region outside their native ranges...”

123-126: I think its important to be clear with the wording: these species haven’t been assessed as “threatened”, they’re inferred to be threatened based on a model. I think this is reasonable to do, its just important to be more clear with the phrasing

RESPONSE 2.7: Because the results reported in the main text are based on *best case scenario* which refers to recent plant extinctions and introductions, and assumes no future extinctions, we have revised the wording to reflect a *best case scenario*, referring to reported recent extinctions, as follows:

*“We demonstrate that changes in α - and β -diversity are driven predominantly by the introduction of non-native species, rather than recent native species extinctions (i.e., under best case scenario of no further extinctions; **Fig. 3**).”*

364-368: These data aggregators should be cited. E.g. GBIF provides download citations.

RESPONSE 2.8: Citation for the data sources have been provided as follows:

- 59. GBIF.org (26 April 2021) GBIF Occurrence Download <https://doi.org/10.15468/dl.7ujp48>
- 60. GBIF.org (8 April 2021) GBIF Occurrence Download <https://doi.org/10.15468/dl.jw4u5a>
- 61. GBIF.org (9 April 2021) GBIF Occurrence Download <https://doi.org/10.15468/dl.m8dzn5>”

368-370: How were the names reconciled? The use of e.g. seems to imply you used multiple taxonomies, but you only list one. If more were listed, it should be made explicit which ones were used. Was this manual cleaning, or did you use some method of automation (e.g. fuzzy matching?) Were records only excluded if they were exact duplicates, or simply identical coordinates? How do you detect uncertain or imprecise coordinates? Did you exclude other common problems (e.g. herbaria, centroids, etc.?)

RESPONSE 2.9: Following the Reviewer's recommendation, we have created a Supplementary material labeled "ODMAP Protocol". In this section, we provide all the methodological details

regarding the generation of species distribution modeling, from input data cleaning to modeled predictions. Specifically for the taxonomic standardization, we clarify that:

We standardized the taxonomy of each species by checking for misspellings, synonyms, formatting errors, hybrid names, and infraspecific ranks, against the backbone taxonomy from the World Flora Online v.2019.05 (WFO 2021). The R package WorldFlora v.1.7 (Kindt 2020) was used to match species names against the static copy (v.2019.05) of the World Flora Online based on direct and fuzzy matching by computing distances between strings following the Levenshtein Distance method.

374: Are the family maps you used included with GreenMaps or published elsewhere online? If not, would be useful to provide these in the SI for reproducibility.

RESPONSE 2.10: We have now provided the family maps as a supplementary information, with range polygons for 413 plant families. This are available as a shapefile of plant families in a folder labeled “APG_FAMILY_MERGED” on the Dryad data repository (<https://datadryad.org/stash/share/UFhi3ts7G6sljHj1IanUUK1V8AVzh4ep6hEUdqCJV9k>).

381: It would be good to specify what the 19 bioclim layers are for those who are unfamiliar.

RESPONSE 2.11: the 19 bioclimatic variables used for the modeling have been included in the supplementary information as follows:

Supplementary Table 1: Bioclimatic variables included in the species distribution modeling of plants of the world.

Variable	Description
BIO1	Annual Mean Temperature
BIO2	Mean Diurnal Range (Mean of monthly (max temp – min temp))
BIO3	Isothermality (BIO2/BIO7) (* 100)
BIO4	Temperature Seasonality (standard deviation *100)
BIO5	Max Temperature of Warmest Month
BIO6	Min Temperature of Coldest Month
BIO7	Temperature Annual Range (BIO5-BIO6)
BIO8	Mean Temperature of Wettest Quarter
BIO9	Mean Temperature of Driest Quarter
BIO10	Mean Temperature of Warmest Quarter

Variable	Description
BIO11	Mean Temperature of Coldest Quarter
BIO12	Annual Precipitation
BIO13	Precipitation of Wettest Month
BIO14	Precipitation of Driest Month
BIO15	Precipitation Seasonality (Coefficient of Variation)
BIO16	Precipitation of Wettest Quarter
BIO17	Precipitation of Driest Quarter
BIO18	Precipitation of Warmest Quarter
BIO19	Precipitation of Coldest Quarter

386: Were maps discarded if they performed poorly?

RESPONSE 2.12: Because our goal is to use species distribution models to generate species maps for use in our analysis, our new R function, *sdm*, was designed with multiple checks such that any species that did not meet one or more checks were filtered out. In addition, the output from the *sdm* function include evaluation statistics in the form of AUC scores or TSS scores for each species. This information is now provided in Lines 428-438 (underlined for emphasis) as follows:

*“We also provide a new R function, sdm, for performing the SDMs across four algorithms (random forest, generalized linear models, gradient boosted machines, and MaxEnt) tailored for SDMs of large datasets. The sdm function is included in our R package *phyloregion*⁶⁷ along with improved documentation and vignettes to show practical application of this functionality under various modelling scenarios. The sdm function was designed with multiple checks such that any species that did not meet one or more checks were filtered out. A feature of novelty of the sdm function is the addition of an algorithm that allows a user to exclude records that occur within a certain distance to herbaria, museums or other infrastructure. By default, we used the most updated version of *Index Herbariorum*, a global directory of herbaria⁶⁸, but a user has the option to specify their own infrastructure to exclude.”*

399-402: I think this is a bit of an overstatement and a bit misleading as phrased. These factors do indeed influence the models, but I think its more appropriate to say these are all things that influence the model (some of which might bias it, e.g. sampling), than things that are captured by it.

RESPONSE 2.13: We appreciate the reviewer's comments. As this version of manuscript was substantially revised to improve clarity and remove redundancy, we have removed discussion on the interpretation of the modeled distributions. It was originally present to set the stage for possible factors that can underlie the predicted species distributions.

410-412: So, for the non-native ranges, were the maps built with ONLY points in the introduced region, or with point in both native and introduced regions? Did you restrict the occurrences to only those that fell within the regions specified by glonaf?

RESPONSE 2.14: Wording has been revised as follows (see also response to comments, above):

“The non-native species ranges were modeled using occurrences that fell outside the boundaries of the native range of each species as determined by Plants of the World Online (POWO). Specifically, we used the following R code to subset occurrences falling outside of POWO as follows:

```
y <- x[!complete.cases(sp::over(x, powo)),]
```

where *x* is a data frame of occurrence of a species, and *powo* a shapefile of the native range of the species. We then used the output *y* to model the distribution of non-native species using the *sdm* function in the R package *phyloregion*⁶⁷.”

483: Unfortunately, recent work by Brody Sandel suggests that this is insufficient to standardize for richness and it is also necessary to rarefy the number of species to a common value and then examine things across many rarefied samples.

RESPONSE 2.15: We assume the reviewer is referring to Sandel (2018) *Ecography* 41: 837–844. This is an interesting and important paper; importantly, however, the bias reported is linked to strongly filtered communities on phylogenetically conserved traits. When these assumptions are relaxed, standard effect sizes adequately control for differences in species richness. Thus we do not believe the results we report here, reflecting shifts in the structure of species assemblages with the addition of non-native species and the loss of threatened species, are unduly influenced. In addition, because we examine both differences in taxonomic richness and standardized phylogenetic diversity, we can reasonably assume that differences in the observed changes in these indices must reflect the phylogenetic component of the latter.

Reviewer #3 (Remarks to the Author):

Comments for the authors

The manuscript entitled “Widespread homogenization of plant communities in the Anthropocene” is based on a study assessing the effects of plant species extinctions and non-native plant species introductions on the homogenization of plant communities globally. To this end the authors model the distribution of more than 200,000 native plant species and nearly 9,000 non-native introduced species, and calculate global diversity indexes under different scenarios. The authors find that non-native species introductions are much more important as drivers of biotic homogenization, compared with species extinctions. This type of studies clearly make a big contribution to our understanding of the impacts of humans on biodiversity. However, I have some concerns that I consider that the authors should address:

- If data on plant species occurrences is not enough to model the distribution for most plant species (as mentioned in lines 139-140 in Daru (2020)), then how can the authors be sure that their analyses based on species distributions are reliable?

Daru, B. H. (2020) GreenMaps: a tool for addressing the Wallacean shortfall in the global distribution of plants, bioRxiv 2020.02.21.960161

RESPONSE 3.1: After data cleaning, we only considered species with at least 20 unique presences. However, we accounted for species with fewer than the threshold number of records for SDMs by using the *bioclim* model in the R package *dismo* to generate initial predictions as additional input occurrence points for downstream modeling. We set the threshold at 0.5 and then used the *randomPoints* function in the R package *dismo* to sample random points based on the suitability model output from the *bioclim* model. We set the *prob* argument in the *randomPoints* function to TRUE, meaning that the values in mask are interpreted as probability weights such that cells with higher suitability will have more probability to be selected as pseudo-presences. These pseudo-points were used in addition to the cleaned dataset as inputs for the species distribution modelling. This information is now provided in the ODMAP protocol in the supplementary material.

- Data on species introductions is likely to be more thorough (showing values closer to reality) than on species extinctions (even while only focusing on recent extinctions), potentially introducing a bias in the analyses presented here. As a result, the authors may be underestimating the effect of species extinctions. I consider that the authors should at least discuss this possibility. This is especially important considering that the authors conclude that the effect of species extinctions is insignificant compared to the effect of non-native species introductions as drivers of biotic homogenization.

RESPONSE 3.2: We thank the reviewer for this suggestion. We have added a discussion about this in Lines 180-191 (underlined for emphasis) as follows:

“Species are not static in their geographic distributions; some may have been moved by people historically, and today many species are tracking shifting climates. We recognize that generating a reliable estimation of the distribution ranges for extinct species would be challenging plus the historical introduction of species beyond their native range may have already contributed to the homogenization of local floras. Likewise, data on species introductions might be more available than data on species extirpations, potentially biasing us to detect a stronger effect of introductions in our analyses. Our analyses thus capture the additional impact on biotic homogenization of more recent anthropogenic activities, and thus likely underestimates the true impact people have on native biodiversity. Further, we believe that as human populations have expanded only relatively recently, historical plant extinctions may have been less likely than historical translocations, and thus our findings that homogenization has been driven largely by introductions, rather than extinctions, is likely conservative.”

- If the authors used GloNAF as a checklist for introduced plant species how is it possible that according to the authors there are 8,839 plant species that have been introduced to a region outside their native range, while GloNAF includes more than 12,000 species (i.e. without considering subspecies and varieties) that have already become naturalized outside their native range? Please explain (Lines 106-107)

RESPONSE 3.3: We agree with the reviewer, and we discuss how we have dealt with this above in response to Reviewer 2's similar comment (see Lines 459-472). Specifically, we used

our new R function *sdm* which has built-in option to restrict predictions within or outside a given boundary, to generate range maps for 10,138 non-native species compared to 8839 previously. All the analyses reported in the revised manuscript are based on the new data.

• Using 1492 as a cut-off date for what is native and what is not native is standard, but may not be correct. There are lots of evidence of pre 1492 movement of species by humans (in Eurasia and South America, are relatively well studied). I suggest the authors make a clearer case on this.

RESPONSE 3.4: We have now added discussion about this caveat in the Introduction in Lines 98-103 as follows (underlined for emphasis):

“We define species composition in the Holocene as the native species’ assemblages in each region before widespread migration by humans as initiated by the Columbian Exchange circa 1492¹⁶. Species composition in the Anthropocene post-date this seminal event, and include non-native introductions, and recent past and projected plant extinctions²⁶. However, there is some evidence of plant introductions by humans across regions in pre-Columbian times^{36,37}.”

Specific comments with line numbers:

Why are there no section titles (Introduction, Results, Discussion)?

RESPONSE 3.5: We have now added section titles in the Results and Discussion following the journal’s formatting guidelines as follows:

“Results and discussion

Temporal changes in α -diversity across plant communities. Under a ‘best case’ scenario defined as recent plant extinctions and introductions,...

Temporal changes in compositional turnover across floras. We found massive global decreases in β -diversity ...

Exchange of non-native plant species and phylogenetic diversity across continents. We additionally illustrate how the exchange of species and phylogenetic diversity between regions is strongly asymmetrical ...

Relationships of non-natives to native flora across spatial scales. Contrary to Darwin’s naturalization hypothesis... ”

Line 54: Please add the following citation:

van Kleunen M, Xu X et al. (2020) Economic use of plants is key to their naturalization success. Nature Communications 11:3201

RESPONSE 3.6: We thank the reviewer for this additional article, which we have included in this revision in Lines 452-453.

Lines 133-134: How can you assess the disproportionate impact of non-native species across dissimilarity metrics from the maps on extended data Figure 3?

RESPONSE 3.7: This is intended as a sensitivity analysis. Given that analysis of beta diversity can be quantified using a variety of approaches including Sorensen, Simpson, UniFrac, Rao’s quadratic entropy, and Jaccard indices, we wished to simply evaluate the robustness of our analysis across the widely used dissimilarity indices such as Simpson, Sorensen and Jaccard.

Regardless of the dissimilarity metric, our results show a general decline in beta diversity over time leading to a homogenization of plant communities in modern times.

Lines 137-140: Where are the results from analyses excluding super invaders?

RESPONSE 3.8: We have now included results from analyses excluding superinvasives in Figure 3 as follows:

Fig. 3 Changes in plant communities under various scenarios of extinctions and introductions in the Anthropocene. Top row a, b, and c shows the differences in α -diversity and bottom row d, and e, shows differences in β -diversity. Comparisons are made across six scenarios: i) 'no extinctions' recent introductions only, ii) 'no superinvasives' based on the removal of non-native species with unusually large invaded ranges, iii) 'Best case' (based on recent extinctions and introductions that have occurred to date), iv) 'business as usual' projected extinction of critically endangered species (CR), v) 'increased extinction' based on projected extinction of endangered (EN) and CR species, and vi) 'worst case' based on projected extinction of all threatened species including vulnerable (VU), EN and CR species. Dashed line at zero corresponds to no change. Species richness was calculated as the numbers of species within 100 km \times 100 km grid cells. Phylogenetic diversity was calculated as the sum of all phylogenetic branch lengths for the set of species within each grid cell.

Lines 147-149: According to Extended data table 2 the principal donors of non-native species are Temperate Asia and North America (not South America).

RESPONSE 3.9: This table has been revised to improve clarity. The table and the Figure 4 are now also better aligned.

Lines 175-177: However, data on species introductions is likely to be more thorough than on species extinctions, potentially introducing a bias in your analyses. As a result, you may be underestimating the effect of species extinctions.

RESPONSE 3.10: This is an important point, and we discuss how we have dealt with this above in response to reviewer's similar comment (Response 3.2) where we added a discussion about this caveat in Lines 180-191 (underlined for emphasis) as follows:

“Species are not static in their geographic distributions; some may have been moved by people historically, and today many species are tracking shifting climates. We recognize that generating a reliable estimation of the distribution ranges for extinct species would be challenging plus the historical introduction of species beyond their native range may have already contributed to the homogenization of local floras. Likewise, data on species introductions might be more available than data on species extirpations, potentially biasing us to detect a stronger effect of introductions in our analyses. Our analyses thus capture the additional impact on biotic homogenization of more recent anthropogenic activities, and thus likely underestimates the true impact people have on native biodiversity. Further, we believe that as human populations have expanded only relatively recently, historical plant extinctions may have been less likely than historical translocations, and thus our findings that homogenization has been driven largely by introductions, rather than extinctions, is likely conservative.”

Line 355: If circles in Figure 4 represent the number of native species in each region does this mean that (according to Figure 4c) the number of native plant species in Europe is notoriously higher than the number of native plant species in South America (including the Amazon rainforest) or in Africa (including the Congo rainforest)? This does not seem to agree with Figure 2 in Daru (2020).

Daru, B. H. (2020) GreenMaps: a tool for addressing the Wallacean shortfall in the global distribution of plants, bioRxiv 2020.02.21.960161

RESPONSE 3.11: We agree with the reviewer, and we discuss how we have revised the figure to improve clarity in response to the first Reviewer's similar comment. For example, the circles represent the number of non-native species or phylogenetic branch lengths exchanged across regions. This revised figure is now provided in Lines 373 (underlined for emphasis) as follows:

Fig. 4 Asymmetrical exchange of phylogenetic diversity and non-native plant species across the world. a, Non-native species originating (outbound arrow) or received (inbound arrows) between each continent. Line thickness is proportional to the number of species exchanged. **b**, Phylogenetic diversity of non-native species originating (outbound arrow) or received (inbound arrows) between each continent. Line thickness is proportional to the sum of branch lengths exchanged. **c**, Net donors and recipients of phylogenetic diversity after correcting for species richness, calculated as the difference in total phylogenetic diversity between the Holocene flora and the Anthropocene flora across continents divided by the number of species exchanged. Arrows indicate the direction of flows from donor to recipient

continent, with line thickness proportional to the sum of shared branch lengths weighted by the inverse of species richness. **The numbers within parenthesis and circle size represents the number of non-native species or phylogenetic branch lengths in each region.** All phylogenetic analyses were ran across 100 trees and obtained a median result. The maps are in Behrmann equal-area projection. A breakdown of nodes and edges exchanged is presented in Supplementary Tables 2–3.

Line 409: I think Richardson et al. (2000b) is a more appropriate citation than the current citation (Richardson et al. 2000a):

Richardson DM, Allsopp N, D'Antonio CM, Milton SJ, Rejmanek M (2000a) Plant invasions—the role of mutualisms. *Biological Reviews* 75:65-93

Richardson DM, Pyšek P, Rejmánek M, Barbour MG, Panetta FD, West CJ (2000b) Naturalization and invasion of alien plants: concepts and definitions. *Diversity and Distributions* 6:93-107

RESPONSE 3.12: We thank the reviewer for this additional article, which we have replaced in this revision in Lines 636-637 as follows:

“71 Richardson, D. M. et al. Naturalization and invasion of alien plants: concepts and definitions. *Divers. Distrib.* 6, 93-107 (2000).”

Line 467: There is a comma missing after “best case”

RESPONSE 3.13: Done.

Peer Review comments, second round review –

Reviewer #1 (Remarks to the Author):

07/10/21 1

The authors appropriately modified the paper according to my suggestions. They are also providing an extensive supplementary material definitively solving one of my points.

I am feeling the ms is now ready for publication.

Compliments for the hard work.

Duccio Rocchini

Reviewer #3 (Remarks to the Author):

The authors have addressed all my comments and the comments from the other reviewers. However, I am still concerned with one of my previous comments:

The authors explicitly use plant introductions and naturalizations interchangeably (author's response 2.6 to reviewer comments), even when these include very different groups of species (Naturalized species are only a small subset of the number of introduced species). I am sure that much more than 10,138 plant species (lines 114-115) have been introduced globally to a region outside their native range. As I mentioned in my previous review, according to GloNAF more than 12,000 species have become naturalized globally, so the number of introduced species must be much higher. If the authors are referring to the number of naturalized species then they should clearly state this because introduced and naturalized species are not the same.

Minor comments:

Line 168: Where does this parenthesis end?

Line 731: Shouldn't it be 10,318 species?

Figure 4: I suggest replacing Northern America with North America, which is how you name this region throughout the text. The same with Southern America

REVIEWERS' COMMENTS

Reviewer #1 (Remarks to the Author):

07/10/21 1

The authors appropriately modified the paper according to my suggestions. They are also providing an extensive supplementary material definitively solving one of my points.

I am feeling the ms is now ready for publication.

Compliments for the hard work.

Duccio Rocchini

RESPONSE 1.1: We thank the reviewer for the positive remarks and for finding our manuscript ready for publication.

Reviewer #3 (Remarks to the Author):

The authors have addressed all my comments and the comments from the other reviewers. However, I am still concerned with one of my previous comments:

RESPONSE 3.1: We thank the reviewer for the positive remarks. In this final version, we have revised the manuscript taking in consideration all of the reviewer's remaining comments.

The authors explicitly use plant introductions and naturalizations interchangeably (author's response 2.6 to reviewer comments), even when these include very different groups of species (Naturalized species are only a small subset of the number of introduced species). I am sure that much more than 10,138 plant species (lines 114-115) have been introduced globally to a region outside their native range. As I mentioned in my previous review, according to GloNAF more than 12,000 species have become naturalized globally, so the number of introduced species must be much higher. If the authors are referring to the number of naturalized species then they should clearly state this because introduced and naturalized species are not the same.

RESPONSE 3.2: We thank the reviewer for this comment. The literature can be rather murky on the usage of these terms, and we acknowledge that we perhaps did not help improve matters. We now refer to naturalized species throughout the manuscript to describe non-native species, including documented records of alien plants that have dispersed into new regions largely by humans, and which have become successfully naturalized. In addition, we have now largely avoided usage of the term "introductions" for example, we have changed "introduced species" to "naturalized species".

Minor comments:

Line 168: Where does this parenthesis end?

RESPONSE 3.3: Checked

Line 731: Shouldn't it be 10,318 species?

RESPONSE 3.4: Corrected

Figure 4: I suggest replacing Northern America with North America, which is how you name this region throughout the text. The same with Southern America

RESPONSE 3.5: We have changed "Northern America" to "North America" and "Southern America" to "South America" throughout the text and figures.